**METHOD**

# INSERT-seq enables high-resolution mapping of genomically integrated DNA using Nanopore sequencing

Dimitrije Ivančić[1,2†], Júlia Mir-Pedrol[1†], Jessica Jaraba-Wallace[1], Núria Rafel[1], Avencia Sanchez-Mejias[1] and Marc Güell[1*]

†Dimitrije Ivančić and Júlia Mir-Pedrol contributed equally to this work.

*Correspondence:
marc.guell@upf.edu

[1] Departament de Medicina i Ciències de la Vida (MELIS), Universitat Pompeu Fabra, Barcelona, Spain
[2] The Barcelona Institute of Science and Technology, Barcelona, Spain

## Abstract

Comprehensive characterisation of genome engineering technologies is relevant for their development and safe use in human gene therapy. Short-read based methods can overlook insertion events in repetitive regions. We develop INSERT-seq, a method that combines targeted amplification of integrated DNA, UMI-based correction of PCR bias and Oxford Nanopore long-read sequencing for robust analysis of DNA integration. The experimental pipeline improves the number of mappable insertions at repetitive regions by 4.8–7.3% and larger repeats are processed with a computational peak calling pipeline. INSERT-seq is a simple, cheap and robust method to quantitatively characterise DNA integration in diverse ex vivo and in vivo samples.

## Background

Development of novel methods for genome editing has enabled an expansion of human gene therapy (HGT) applications [1, 2]. Many HGT strategies are based on the addition of a DNA payload that can compensate for a genetic defect and/or provide the recipient cells with a synthetic feature [3]. Delivery of therapeutic payloads has traditionally been based on both viral and non-viral vectors that uncontrollably integrate across the genome. Whilst precise delivery technologies are emerging [1, 2], viral vector mediated uncontrolled delivery has benefited an important number of patients [4]. Insertional mutagenesis events have been observed, halting first attempts of gene therapy trials after cases of leukaemia were linked to a retrovirus-based SCID-X1 gene therapy [5]. Vector-driven clonal expansions have been observed again recently in CAR-T therapy [6, 7] and in a leukaemia case linked to the retroviral based drug Strimvelis [8]. rAAV can cause insertional mutagenesis and induce clonal expansions [9, 10]. Viral-free technologies such as the PiggyBac transposase (PB) have been linked to oncogenic induction [11].

The discovery and implementation of precise delivery strategies using tools such as ZFN, TALENs and CRISPR-Cas9 [12, 13] can potentially avoid unintended gene disruption or activation and perform simultaneous gene inactivation and addition, which are key features for therapeutic genome modulation. Such techniques have also raised concerns related to safety; It has been reported that the integration of payloads can be imprecise, resulting in different unexpected integration outcomes, like integration of donors at off-target cut sites and genomic rearrangements such as inversions or translocations [14]. In both uncontrolled and precise gene delivery strategies, comprehensive characterisation of integration events is critical for evaluating safety and precision. Experimental procedures have been reported for the characterisation of Lentiviral (LV) vectors [15–17], rAAV vectors [9, 18, 19], Transposon delivery [20] or CRISPR/cas9 mediated integration [14], relying on distinct methods such as Lam-PCR [21] and nr-LAM-PCR [22], GUIDE-seq [23] and TLA [24]. Moreover, insertional profiling can reveal vector feature integration preferences [25], which can be used for clonal tracking of edited cells and for inference of clonal expansions due to genome vandalism [15, 26, 27]). Insertion signatures can also be used for diagnostic purposes [28] and nuclease off-target profiling in methods that rely on a dsODN [29] or LV vector capture in Double Strand Breaks [30]. Off note, the above described examples rely on short read technologies, which, due to the repetitive nature of the human genome, can potentially overlook certain editing outcomes, and clonal expansion of integrated DNA in repetitive regions has been reported [20, 27]. Long read sequencing (LRS) technologies can generate reads of kilobases in length, being suitable for the determination of repeat sequences, most notably for the centromeres [31, 32] or detection of structural variants [33, 34]. To study the effect of repetitive elements on capturing insertion sites, we implemented a model that showed significant dependency of read length for accurately resolving insertion sites. Next, we developed INSERT-seq, an insertion detection workflow consisting in a single-tail adapter/tag (STAT-PCR) library prep and oxford nanopore long read sequencing. We complement the experimental protocol compatible with longer read length with a computational peak calling pipeline implemented in nextflow and available through a web server. We tested INSERT-seq for LV, rAAV, and PiggyBac payload insertions, in both cell lines and mouse derived samples. INSERT-seq can be used to resolve unknown integration sites of a payload in an edited genome with 1% detection limit in a robust, fast, and low-cost manner.

## Results

### Read length dependency

To determine the resolution gain that increased read length junction capture could provide, we generated a simulated dataset with increasing read length of the insertion-genome junction (Fig. 1a). We sampled random positions from the reference genome and generated reads around these positions of 250 and 1000 bp in length. A 3.9% increase in overall detected insertion sites was detected when size was increased from 250 to 1000 bp (Fig. 1a). We observed that the junctions skipped by short reads laying in repetitive regions belong to the longest mobile genetic elements in the genome, mostly in LTR retrotransposons, LINEs and SINEs where 4, 3 and 2 new insertions were detected respectively with long-reads (Fig. 1b). Furthermore,

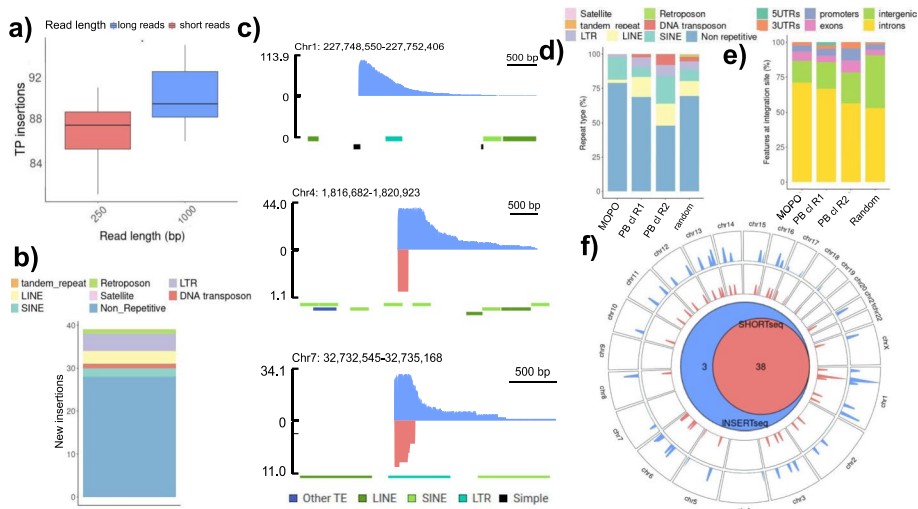

**Fig. 1** Read-length impact in resolving insertion sites. **a** Number of true positive (TP) insertions detected in a simulated dataset of 100 random insertions and 10 replicates, with reads of either 250 bp and 1000 bp. Significance *t*-test (*$p < 0.05$). **b** Number of new insertions detected with long reads (1000 bp) compared to short reads (250 bp) from the simulation of a total of 1000 random insertions (**a**) and the repetitive region where they land. An insertion was considered within a repetitive region if the insertion site $\pm$ 10 base pairs were falling inside a repetitive region from Dfam database. **c** Coverage at selected lentiviral insertion sites in the mono-clonal poli-insertional (MOPO) analysed cell line, with either short (red) or long (blue) sequencing technologies. Repetitive regions from Dfam database human genome hg38 are annotated. The three new insertions detected only with INSERTseq are shown. **d** Number of lentiviral and transposase insertions falling in a repetitive region and repeat type in MOPO sample and "100 PB clones" sample, respectively. One hundred clones sample is originated from expanding 100 cells in which stable integration of a PB transposon has occurred (~100 integrations expected). **e** Genomic features at integration sites in MOPO sample and "100 PB clones" sample. **f** Genome-wide map displaying the overlap between insertions detected in MOPO with either short read insertion detection (red) and long read insertion detection (blue). The Venn diagram summarises the overlap of common insertions with the two methods

examining the length of all repetitive regions from the human genome annotated in Dfam database [35], we detected a high number of repeats not resolvable by short read sequencing (longer than 500 bp) (Additional file 1: Fig. 1). With simulated reads of 1000 bp, there were still 101 of 1000 insertions unidentified (Table S3). Those correspond to reads that map to multiple regions in the reference genome, even though those regions are not annotated to contain repetitive regions. Fourteen of those 101 insertions are detected when read length is increased to 5000 bp (Additional file 1: Fig. S8d), suggesting that the sensitivity of the method increases when read length increases. We analysed the impact of sequencing output to detecting integration sites (Additional file 1: Fig. S8c). We found that there was no increase in detection upon increasing bp output starting from 250Mbp to 1.5Gbp for the simulated insertions.

To implement a long-read based library prep procedure in the optimal range predicted by the model (~1.5kb, Additional file 1: Fig. S1), we used a nested single-tail adapter/tag (STAT-PCR) [29] coupled to nanopore sequencing to obtain long read insertion-junction capture (Fig. 2a). STAT-PCR single tailed adaptor ensures selective amplification of fragments containing both adaptor and vector sequence, because the primer targeting the adaptor region does not bind to the single stranded version of the adapter, and it only binds when amplification from the primer targeting the vector

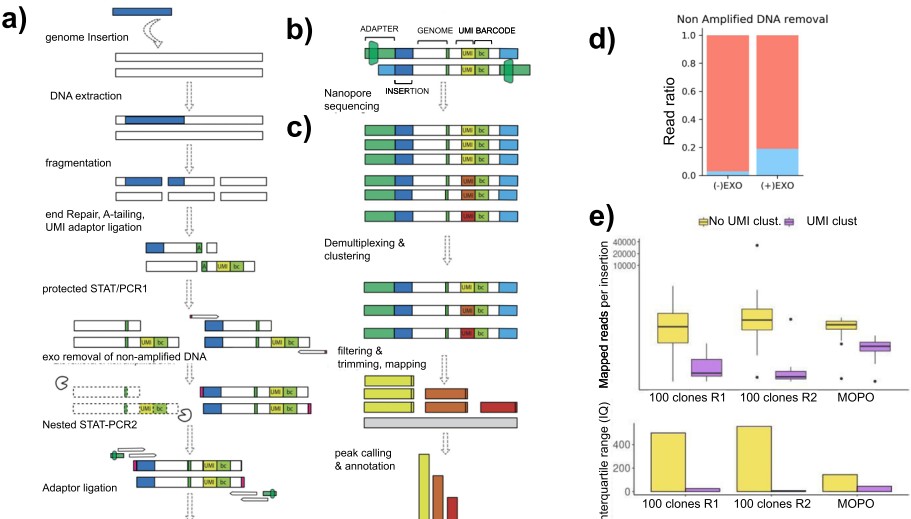

**Fig. 2** Implementation of the optimised INSERTseq protocol. **a** Overview of the library prep process. Genomic DNA is extracted, fragmented, end-repaired and A-tailed followed by ligation of an adaptor that contains an UMI for read clustering and a barcode for sample demultiplexing in the computational pipeline. **b** Schematic representation of sequenced reads structure. **c** Overview of the analysis pipeline. Reads are clustered by UMI, integration sequence and adapters are filtered and trimmed, reads are mapped against the reference genome and significant peaks are reported and annotated. **d** Exo I endonuclease effect on fraction of enriched reads, represented as fraction of LTR enriched reads (blue) over non-enriched. **e** Comparison of count distribution for each insertion with (purple) or without (yellow) UMI clustering in MOPO sample and "100 PB clones" sample. Top panel shows the distribution of the number of reads per insertion in logarithmic scale. Bottom panel shows the interquartile range of the distribution

occurs. Comparing nanopore-based INSERT-seq to short read based platforms on a clonally expanded isogenic cell line containing multiple lentiviral insertions (MOPO sample) (Additional file 1: Fig. S2) revealed a gain in junction detection. Three new insertions (7.3%) were detected (Fig. 1c, 1f, Table S1: Table S4). Newly detected insertion at chr1 falls in a simple repeat, the one in chr4 falls in a region with multiple SINEs and the one in chr7 in an LTR. Short reads landing in these regions are unable to map properly to the reference or are removed during the filtering step due to the quality of mapping. A comparison with higher short read count was made (Additional file 1: Fig. S8a) where one insertion at chr7 is recovered (Additional file 1: Fig. S8b). The increase of these two new insertions detected with long reads in contrast to short reads, from a total of 41 insertions, is of 4.8%. Since the library protocol of INSERTseq is amplification based, the average read length decreases after library prep compared to fragmented DNA. Mapped read length of long reads ranges from 41 to 5718bp with a mean of 329bp whilst mapped length of short reads range from 25 to 239 bp with a mean of 109bp in the analysed MOPO sample (Additional file 1: Fig. S10). The increase of detected insertions was analysed with modelled reads of 100 (short) and 350 (long) base pairs, finding an increase of 2% (Additional file 1: Fig. S8e). Such results confirm that the potential of INSERTseq improves with longer reads.

In order to validate the reproducibility of the method, we repeated the measurement on the MOPO sample with a different 5′ primer, and all 41 insertions where detected (Additional file 1: Fig. S9). We also captured the 3′, detecting two new integration sites (Additional file 1: Fig. S9). Two insertions detected in the 5′ sample were not detected by

the 3′ mapping (Additional file 1: Fig. S9). The insertions found only with the 3′ end mapping correspond to an insertion at chr10 falling in an LTR (MLT1C2) and one at chr22 falling in a region with multiple repeats classified as unknown in the Dfam database. The two insertions found only with the 5′ end mapping correspond to the insertion at chr7 which was also not detected with SHORT-seq and an insertion at chr1 falling in a SINE (AliSz6). In the analysed MOPO sample, the insertions found in repetitive regions comprised LINEs, SINEs, LTR retrotransposons and DNA transposons (Fig. 1d) and insertions were happening preferentially in intronic regions with a higher percentage than the randomly distributed model sample (Fig. 1e). Importantly, LV integrations used in this experimental model can have site selection biases [25]. Changes in abundance of repetitive elements in the vector preferred regions might render differences in the overall detection of insertion sites, making the impact of read length in resolving site selection variable for each vector (Fig. 1d). From the 3 insertions not detected with SHORT-seq, one was located in a simple repeat which is difficult to sequence by Illumina technologies and three of them contained repetitive regions in the sequencing range.

To detect unmappable insertions in repetitive regions, a pipeline for unanchored peak calling was implemented to allow the detection of insertions happening within repetitive regions in the genome (Additional file 1: Fig. S4a-b). We were able to call 10 different unanchored insertions in the MOPO clonal cell line. Four peaks occur in MER74A repeats, two in MER74B, and in MER133A, MLT1C2, MER74C and HERVL11 one (Additional file 1: Fig. S4c). When applying the unanchored peak calling to SHORT-seq data, two peaks are found at MLT1C2, a mammalian-specific non-autonomous LTR element dependent on class 3 ERVL which was also found by the unanchored peak calling algorithm in INSERT-seq data, and LTR33A, a class 3 endogenous retrovirus with HERVL. Furthermore, one of the insertions that was able to be detected only when mapping the 3′ end of the integrated lentivirus happened at an MLT1C2 repeat, which was detected by the unanchored peak calling. However, due to the technical challenges that repetitive regions present, unanchored peaks are only reported as putative peaks.

### Overview of library prep implementation and optimisation

Upon collection of the sample of interest, genomic DNA from an edited cell population is shredded using a g-TUBE column and an adaptor containing a barcode for demultiplexing and an unimolecular identifier (UMI) is ligated. Genome-junctions are amplified with two nested STAT-PCRs [29]. UMIs are subsequently used to remove PCR duplicates after amplification (Fig 2a, b). The use of UMIs has been used for the correction of PCR artefacts [36], avoiding the overrepresentation of certain insertions and thus allowing better quantification of insertion sites (Fig. 2e). UMI clustering can correct insertion site count, as seen for an insertion profile of a population of 100 clones edited with HypB transposase (referred to as PB clones) (Fig. 2e). In order to increase the ratio of genome-junction reads over non-amplified, non-specific reads, lambda exonuclease is used to digest the non-amplified DNA present in the sample after the first STAT-PCR. Primers with 5′ phosphorothioate bonds protect amplified DNA from digestion. Adding phosphorothioate bonds inhibits lambda activity and increases total output of enriched reads with junction (Fig. 2d). To measure limit of detection (LOD), a mono-clonal cell line with one targeted insertion in chr12 (MN2)

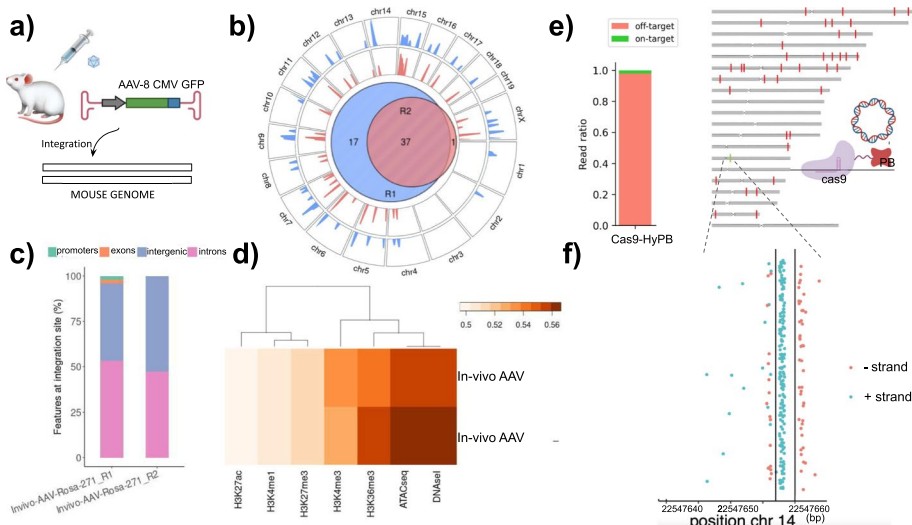

**Fig. 3** INSERTseq reveals integration patterns in rAAV-8 and PB genome editing frameworks. analysis reveals rAAV-8 integration hotspots in mice. **a** AAV-8 CMV GFP was administered to mice using retro-orbital injection, and liver tissue was extracted for INSERTseq analysis 1 week upon administration. **b** Genome-wide map displaying integrations of rAAV in two replicate samples. The Venn diagram summarises the overlap of common insertions of both replicates. **c** Features at AAVs integration sites. **d** Heatmap of the area under the curve (AUC) representing the association of integration of AAV mouse liver samples (replicate 1 and 2) with different features. An AUC value higher than 0.5 indicates positive association (more than random) with the feature, whilst an AUC lower than 0.5 indicates negative association. **e** Genome wide integration of a Cas9-HyPB transposase, representing off-target integration (red) and on-target integration (green) at the TRAC gene (ch:r1422,547,659). Fraction of on-target (0.016) and off-target (0.98) insertions is displayed in a stacked barplot. **f** Dotplot displays on-target integration events in both + strand (blue dots) and − strand (red dots). Vertical lines represent region targeted by gRNA (outter black lines) and Cas9 cutting site (middle black line)

was diluted with a constant amount of a different monoclonal cell line with one insertion at chr6 (MN7) as a reference. To validate the results, the same procedure was repeated by diluting a different monoclonal cell line with two targeted insertions at chr3 and chr9 (MN3) with MN2. The serial dilutions were 1:1, 1:10, 1:100, 1:1000 and 1:10000 The detection of the target integration site at the dilution 1:100 and not at the dilution 1:1000 and 1:10000 determines INSERT-seq limit of detection of 1% (Additional file 1: Fig. S3).

### rAAV-8 genome wide integration in mice liver

To evaluate INSERT-seq performance in an in-vivo model, liver tissue from mice administered with rAAV serotype 8 was studied to determine the genome-wide integration profile of rAAV. rAAV vectors are being widely tested in gene therapy trials [37] and their genomic integration has been associated with hepatocellular carcinoma in mice [10] and clonal expansion upon long term administration in dogs [9]. Recurrent integration sites have been previously detected in mouse brain [23] and liver [38] and hotspot differences across tissues have been reported [19]. We were able to detect 55 insertions, 37 of them common between two replicates (Fig. 3b). We observed characteristic auto-integration of rAAV genome in itself (Additional file 1: Fig. S5a).

We also observed a preferential insertion of rAAV at intronic regions (Fig. 3c) and an association of insertion sites to open chromatin regions (compared with ATACseq and DNAseI datasets) (Fig. 3d).

### Assessment of a Cas9-PiggyBac chimeric transposase

To assess potential of INSERT-seq for analysing emerging precise gene delivery vectors, we determined integration sites in a Cas9-PiggyBac chimeric fusion previously reported [39]. We detected insertion at the Cas9 induced double-strand-break in the TRAC locus, which was nevertheless very low compared to total integration with on-target fraction being 0.016 and off-target fraction being 0.98 (Fig. 3e, f). Several off-target insertions had high coverage and seemed to originate from a single insertion (Additional file 1: Fig. S6, Table S1). We suspect that this could be due to positive selection of the clone harbouring insertion in this particular site, due to the time passed from transposon transfection (~3 weeks). Insertion site determination in cells treated with Cas9-HyPbase in shorter periods of time (~1 week) resulted in very high capture of transposon plasmid (corresponding to episomal DNA) (data not shown), substantially reducing coverage of payload-genome junctions. Adding a PCR blocking oligo at the plasmid flanking region (described in [16]) could be a solution for increasing coverage of insertion sites in recently transfected cells.

### Discussion

Repetitive regions vary in abundance and are usually between 100 bp and 5000 bp [40] (Additional file 1: Fig. S1b). The use of Oxford Nanopore sequencing generates longer reads (~2 kb), which allow the detection of integration sites overlooked by short read sequencing methods, providing between 4.8 and 7.3% improvement in insertion site detection sites landing in repetitive regions. Although our estimates are measured in lentiviral vector integrations, differential integration site preferences across vectors [25] might in turn affect the percentage of reads landing in repetitive regions. INSERT-seq library preparation is based on two nested STAT-PCRs [29] to capture integration-genome junctions and an exonuclease step between the two nested PCRs to remove the non-amplified reads. Incorporation and subsequent clustering of UMI containing adaptors allows for correction of PCR biases, making INSERT-seq a suitable tool for quantification of on-target vs off-target integrations. We demonstrated this by the characterisation of the integration profile of a Cas9 protein fused to HyPbase which revealed that on-target integration occurs but in a small proportion, suggesting that fusion of the both proteins is not enough to obtain efficient programmable insertion activity. The calculation of the LOD shows that INSERT-seq can sensitively detect insertion sites with frequencies as low as 1%. Such sensitivity could be improved with more sequencing depth. Careful measures to avoid contamination at the library prep steps and barcode addition prior to amplification is critical to avoid cross contamination of samples analysed in the same batch, especially in low enriched samples. We have also been able to develop a computational method to detect unanchored integrations happening at repetitive regions, which are undetectable by the standard analysis. Pipeline Implementation through Nextflow and Docker containers ensures easy installation, handling and high reproducibility. INSERT-seq can be adapted to capture canonical integration of

any DNA payload across a genome, and improved detection sensitivity provided by this method can provide improved detection of genome vandalism events and non-canonical integration outcomes if combined with primer tiling [41]. With the current protocol however, unbiased detection of partial payload integrations in which ends are missing, previously reported for rAAVs [41], remains unsolved. Implementing INSERT-seq to characterise novel targeted delivery strategies such as PEs [42], PASTE [43] and targeted integrases can robustly assess precision of these tools. Protocol implementation should be feasible across multiple organisms as novel genome engineering technologies arise.

## Conclusions

INSERT-seq incorporates amplification based enrichment and UMI amplification correction with a computational pipeline to process integration sites. It can resolve DNA integration sites with increased resolution compared to traditional short read sequencing methods providing higher accuracy in determining integrations in repetitive regions, which compose an important subset of many eukaryotic genomes. We have successfully used INSERT-seq method to analyse lentiviral, transposon and rAAV DNA payloads in the genome of both in vitro and in vivo samples. Reduced hardware requirement in Oxford Nanopore Technologies (ONT) platform and sample processing time should make the method practical across different research environments, as for example in clinical settings, and INSERT-seq should be applicable to other long read sequencing platforms such as PacBio. The protocol is available in the additional material and the code for analysis at Bibucket and as a web application (http://synbio.upf.edu/insertseq/).

## Methods

### Read length impact model

In order to assess the impact of read length in the detection of insertions, we performed a simulation of data using different read lengths. Reads of 250 and 1000 bases were generated for a total of 1000 random genomic locations with R scripting and Biostrings [44]. We performed 10 replicates, simulating 100 insertions for each condition. A normal distribution was applied to read length, setting the standard deviation to 10 for short reads (250 bp) and 25 for long reads (1000 bp). The human genome hg38 was used as reference. An unmasked genome was used to allow the presence of repetitive regions in the obtained reads. A total of 1 million reads was generated for each length. The mapping of reads was performed following the analysis pipeline steps (see below). Peak calling was not applied. The human genome (hg38) was used for the analysis. An insertion is considered to fall in a repetitive region if the insertion site or 20 base pairs surrounding it contained a repetitive region from Dfam database. The same procedure was repeated for read lengths of 100, 350 and 5000 bases for further comparisons.

HEK293T cells [ATCC® ACS-4500™] were grown at 37° and 5% $CO_2$, and Dulbecco's modified eagle medium high glucose (DMEM) (Gibco, Thermo Fisher), 10% FBS, 2 mM glutamine, 0.1 mg/mLpenicillin and streptomycin, was used. K-562 cells [ATCC® ACS-CCL-243™] were grown in RPMI 1640 (Gibco, Thermo Fisher) supplemented with 10% FBS. To generate MC-2 cell line with isogenic insertional signature, HEK293T cells were transduced with pSICO LV vector at MOI 1, and GFP fluorescent single cells were isolated in to 96 well plates using BD FACSAria (Biosciences) cell sorter and further

expanded and validated for GFP expression with BD LSR Fortessa cytometer (BD Biosciences). Mono-clonal poli-insertional cell line (MOPO) was generated following the above described procedure transducing cells at MOI ~40 (Additional file 1: Fig. S2). MC samples from Supplementary Fig. 3 were generated following the same procedure but with cells infected at MOI 1, mixing a constant amount of one cell line with different amounts of a different cell line. "100 clones PB sample" was generated by transfecting HEK293T cells with HyPbase and PB-512 (transposon) plasmids, or HyPbase-Cas9 fusion, gRNA AAVS1-3 and PB-512 (transposon) and sorting 100 cells 2 weeks after transfection (~100 integrations expected). HyPB-Cas9 sample from Fig. 3e was generated following the same procedure as for "100 clones PB sample" but with plasmids HyPbase-Cas9 fusion, gRNA AAVS1-3 and PB-512 (transposon). pSICO LV vector was generated by packaging pSico plasmid, as a gift from Tyler Jacks (Addgene plasmid # 11578) following previously described protocol [45]. qPCR was performed using dsDNA Dye and probes for RNaseP and psi sequences (Table S2).

### rAAV8 administration in WT mice

rAAV8 GFP viral vectors were purchased from the Viral Vector Production Unit (UPV) at Universitat Autonoma of Barcelona. 5*10E+11rAAV8 GFP particles were retro-orbitally injected in Rosa26-Cas9 knockin on B6J (JAX stock #028555) 8 weeks old mice. Nine days after rAAV8 administration, mice were sacrificed and liver tissue was extracted and homogenised. Animals were purchased from Jackson Laboratories; male and female were used without distinction.

### Library preparation and sequencing

For SHORT-seq implementation, DNA was extracted using DNeasy Blood and tissue kit (Qiagen) and fragmented to ~ 300bp fragments on a Q800R3 Sonicator using following parameters: 40% Amplitude, 15/15 Pulse, and 1.25 min total sonication time. End repair, A-tailing, and ligation of Y-adapter [KAPA Hyper Prep Kit (KR0961 – v5.16)] were performed with 5µg of fragmented DNA, followed by SPRI bead purification using x1 bead ratio. Two nested single-tail adapter/tag (STAT)-PCR [29] were performed with LongAmp Hot Start Taq polymerase (NEB, M0533S) using adaptor binding P5_1 and P5_2 primers and corresponding insert primer pairs (see Table S2). For long read INSERT-seq implementation, DNA was extracted using Nanobind kit (Circulomics, catalogue no. NB-900-001-01), sheared to ~6 kbp fragments using g-TUBE™ (Covaris, catalogue no. 520079). WGP primer mix from Nanopore PCR Barcoding Kit (SQK-PBK004) was additionally added to the second PCR. Sequencing was performed in Flongle R9.4.1 flowcells obtaining a total output of ~ 300000 reads. For the calculation of the limit of detection (LOD), a mono-clonal sample from HEK293T cells with one true lentiviral insertion was diluted with another monoclonal cell line at the proportions 1/100, 1/100, 1/1000 and 1/10000. Dilutions were sequenced a Flongle R9.4.1 flowcell obtaining a sequencing output of ~8M reads in total. The analysis was performed following the INSERT-seq analysis pipeline detailed in the Methods section.

**Integration site analysis**

Nanopore raw reads were basecalled using Guppy 4.0.11 (made available by ONT via their webpage https://community.nanoporetech.com); read quality was assessed with NanoStats NanoQC and NanoPlot from NanoPack [46]. Reads were filtered by quality (> 10) and length (> 200) with NanoFilt from NanoPack.

Reads were clustered by UMI using a combination and adaptation of two previously published pipelines (pipeline-umi-amplicon distributed by ONT https://github.com/nanoporetech/pipeline-umi-amplicon and longread_umi [36]). Briefly, the clustering was performed by extracting the UMI sequences with Python scripting, sequences were clustered with vsearch [47], and the consensus sequence of the clusters was obtained by performing two rounds of polishing with racon [48] and two rounds of medaka (https://github.com/nanoporetech/medaka).

For the analysis of insertions (Fig. 2c), reads were filtered to force the presence of used adapters and trimmed to remove the adapter and insert sequence from the read with cutadapt [49]. Reads were mapped against the reference genome with minimap2 [50] "map-ont" default parameters and filtered with Python scripting, selecting uniquely mapping reads with a map quality higher than 30. A first output is returned with bedtools [51] in bed format containing all mapped reads. Afterwards, a peak calling step is performed with Python scripting where peaks are filtered by shape. A peak is considered to pass the shape filter when the Residuals Sum of Squares (RSS) of fitting the peak coverage to a beta distribution is lower than 1.

A set of 42 manually selected true insertions and 3393 negative insertions was used to assess the RSS threshold of 1 for peak shape filtering. (Additional file 1: Fig. S7).

The INSERT-seq pipeline has been implemented with Nextflow [52] and Docker [53] containerisation to allow an easy and reproducible analysis of the results.

llumina short reads of 250 bp were clustered by UMI following the pipeline from Tsai et. al. (2015) [29] subsampled to obtain the same coverage as with long reads (~30K total reads). T[29] reads were mapped against the reference genome with minimap2 "sr" default parameters and "--secondary=yes" and filtered selecting uniquely mapping reads with a map quality higher than 30. A peak calling shape filtering step is performed with a RSS threshold of 1.The analysis was repeated for the complete ~230000 consensus sequences after UMI clustering. In order to analyse the association of integration with different features, 10000 random positions were obtained from the reference mouse genome mm10 as negative controls. Positive insertion positions were obtained from in vivo experiments. Different feature annotations were obtained from ENCODE liver mouse (mm10) stage P0; the analysed features were H3K9me3 (ENCFF166MIE), H3K9ac (ENCFF331CRG), H3K27ac (ENCFF535FJV), H3K27me3 (ENCFF321SJE), H3K4me1 (ENCFF-969HOM), H3K4me2 (ENCFF123DZJ), H3K4me3 (ENCFF612SAB), H3K36me3 (ENCFF361EKE), DNAseI (ENCSR216UMD) and ATACseq (ENCFF764NTQ). The number of peaks was counted within 100 kilobase pairs surrounding each insertion site. A ROC curve was computed for each feature and the area under the curve (AUC) was plotted in a heatmap.

### Peak calling of repetitive regions

All the reads that do not map against the reference genome have secondary alignments or map with a quality lower than 30 are labelled as unanchored reads. Those potentially belong to an insertion happening in a repetitive region in the genome (Additional file 1: Fig. S4a).

In order to call insertions in repetitive regions, unanchored reads were mapped against the reference sequence of the true detected insertions with minimap2 to discard those that belong to already called insertions. Afterwards, selected reads are classified to a repeat based on Hidden Markov Model (HMM) hits obtained from Dfam database [54]. The reads from each repeat are clustered with vsearch [47] to determine if all the reads belong to the same peak. Finally, a coverage filter of 50% the mean coverage of called insertions and shape filter are applied to each cluster in order to call an unanchored peak (Additional file 1: Fig. S4b).

For short reads, the same algorithm was applied with modified mapping parameters to "sr".

## Supplementary Information

---

Additional file 1: Supplementary figures 1-11 and INSERT-seq protocol.

Additional file 2: Supplementary tables S1-S4.

Additional file 3. Review history.

---

**Peer review information**

**Review history**

The review history is available as Additional file 3.

**Authors' contributions**

MG conceived the study. MG, DI and JM designed the experiments. DI and JJW performed the experiments with help from NR. JJW performed mice experiments with help from ASM. JM and DI analysed the data. JM wrote the code and the INSERT-seq pipeline. MG, JM and DI wrote the manuscript with input from all authors. The authors read and approved the final manuscript.

**Funding**

We thank funding received from UPGRADE (European Union Horizon 2020, grant agreement No 825825), Fundación Ramón Areces ("Advanced gene editing technologies to restore LAMA2 on merosin-deficient congenital muscular dystrophy type 1A"), MdM projecte de recerca "Unidad de Excelencia María de Maeztu", funded by the AEI (CEX2018-000792-M).

**Availability of data and materials**

Next-generation sequencing data are available in the European Nucleotide Archive under the Study accession number PRJEB46760 [55].

Code for INSERT-seq pipeline has been made available in Bitbucket https://bitbucket.org/synbiolab/INSERT-seq_pipeline/ [56] under the MIT license and through the web page application synbio.upf.edu/INSERT-seq/. The source code has been deposited at Zenodo under the DOI 10.5281/zenodo.7114314 [57]. This pipeline will also be added to the NF-core community.

## Declarations

**Ethics approval and consent to participate**

All animal procedures were approved by the Animal Experimentation ethics committee of Barcelona Biomedical Research Park (protocol number MGC2-19-0033-P2).

**Competing interests**

The authors declare no competing interests.

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

## 