## [Additional file 3. Review history. · Genome Biology]

Review History

First round of review

Reviewer 1

Were you able to assess all statistics in the manuscript, including the appropriateness of statistical tests used? No.

Were you able to directly test the methods? No.

Comments to author:

INSERT-seq enables high resolution mapping of genomically integrated DNA using single molecule long read technologies written by Dimitrije Ivancic and co-workers.

Viral vector mediated delivery is a method currently widely adopted in the field of human therapy applications. As recently examples were published of gene therapy trials based on such methods leading to undesired side effects the need for methods that can control for correct, but also undesired effects are of high interest.

This manuscript describes the development and initial validation of the INSERT-seq method. This method promises for the sensitive detection of LTR mediated DNA insertions at a-priory unknown positions and is, given the above, most relevant and timely.

Technically this method is based on the principle of STAT-seq/GUIDE-seq except here long read sequencing (LRS) is used yielding an increase in the number of uniquely identifiable integration sites. In addition, the authors adopted a mapping strategy designed to deal with integration sites yielding non-unique sequence reads.

The authors introduce INSERT-seq having performed a first set of validation experiments making it highly likely that their method outperforms STAT-PCR.

However, presented data provides insufficient body to support the claims as currently presented. Also, the manuscript looks to be written hastily leaving many incorrect referrals, missing legends figure numbers, performed but not described experiments making it difficult to read through in general. It will take serious effort correcting this. However, as most issues are textual, and few additional experiments are required I would recommend considering this manuscript for publication when corrections have been made.

Specific comments in order of appearance

1. Title is too generic for the content of the manuscript

a. Due to placement of primer towards integration site in its current form only applicable to DNA that insert via known sequences like LTRs.

b. The title states 'long read technologies' as in plural, except a single method was used for validation.

2. In figure 1a the authors aim to provide evidence for the benefit of using long read sequencing to increase the uniqueness of the identified integration site sequences. Hereto they use 100 random positions in the genome and make a comparison between a 250 and a 1000bp sequence at that position as described in the methods section. From that experiment INSERT-seq is

claimed to provide a 4.2% benefit in the ability of uniquely calling integration site. This experiment raises three questions currently unaddressed

a. For this computational analysis to be relevant to the remainder of the manuscript, and as the use of long-read sequencing icw STAT-PCR is presented as novelty it is important to verify that the assumed 1kb read length matches the length of the experimental data. Aiming to verify this parameter from the current manuscript the majority of NGS reads are within 200bp from the insertion site (Figure 1c and supplemental figure 8) it will be good to address this properly. In addition, a Covaris G-tube was used to produce 2kb fragments, while these tubes are meant to produce 6-12kb fragments according to specifications. Currently it is not clear how the Covaris protocol was adjusted to yield 2kb fragments.

b. In figure 1b the finding of close to 40 new integrations in this dataset used to prepare figure 1a, of which in main text it is mentioned that 4.2% more sites are detected while in the methods this computational experiment was performed on 100 randomly selected sites. These number does not add up and suggest that more than 100 sites were used for this analysis. Please verify.

c. The total number of identified sites does not add up to 100 (in total or % is unclear from figure 1a), as at most 90 are found; all methods have their limitations and it is important to understand what these are; can the authors comment on what sites they did not identify and what was the cause?

3. In both the abstract and discussion a resolution improvement of 10% is claimed as a general performance increase for INSERT-seq. Though the origin of the 10% is not explained, to my knowledge the only experiment showing nearing this figure is the experiment shown in figure 1f. This single experiment covering ~40 integrations is however not sufficient to make such a broad claim. For example, from the computational experiment the increase is shown to be 4.2% which is quite a different range and interrogating many more genomic locations.

4. For the purpose the authors generated the MOPO control cell line to be used as a reference. In the main text pg 3 line ~15 it states that 42 viral insertions should be present. Supplemental figure 2 however shows 37 copies to be present. That finding makes that the reference value looks more alike the short-read finding which is presented as being incomplete. I'm afraid that with this outcome of the ddPCR (here used as the independent reference method) the authors should make an extra effort on showing the additional sites are indeed true positives as now presented.

5. While introducing a new quality control method no replicate experiments are shown. The MOPO and possibly the 100 clones PB1 sample would make good candidates.

6. Most of the data presented in figure 1 d and e is not described in the main text.

7. When describing the limit of detection it is currently not clear how this number was obtained. In supplemental figure 3 coverage and DHD are plotted vs proportion positive sample (%), but how many reads were used or how peak calling was executed including significance cut-of percentages and how the increased number of false insertion sites is treated in these experiments is not explained.

8. Supplemental figure 6c is difficult to understand without figure legends or explanation.

9. Out of the 4 new sites presented in figure 1c. It is not clear

a. why the top 2 integration sites do not show any reads using short read-seq while the bottom two show very high coverage and

b. having such high coverage, why are the bottom two peaks missed in peak calling?

10. In general it was difficult to read through the manuscript due to

a. the lack of page numbers, (for reference below, pg1 is considered the title/abstract pg)

- b. the lack of figure numbers
- c. the lack of figure legends.
- d. Pg1 line 40; 'in-vivo' where ex-vivo is the more correct description.
- e. Pg2 line 24: STAT-PCR [23] was called GUIDE-seq by the authors of the original publication
- f. Pg2 line 44: no explanation of 'PB' which can easily be done in line 23 of the same document
- g. Pg 3 line 8-9 unclear what is meant with '..we detect a high number of repeats (supplementary Fig. 1).' How does the sequence length impact on the length of a repeat?
- h. Pg3 line 11 or 12 (line numbering does not align with the actual text) it states '... in the optimal range predicted by the model.' However, the mentioned model is not explained nor introduced.
- i. The y-axis on figure 2e top panel is incomplete
- j. pg3 line 50. It is not clear where the referral of '...one overrepresented insertion....(Fig. 2e). refers to when looking at figure 2e.
- k. pg3 line 60 (bottom line) supl figure 3c does not exist, please refer to correct figure
- l. pg4 line 19: mentioning of 'a clinically relevant sample' followed by 'liver tissue from mice' raises a question mark. Though one can argue that such an animal experiment is related to the direct medical treatment or testing of patients, but it reads artificial/farfetched as the relationship is indirect at most. Rephrasing this sentence would help to increase credibility.

Reviewer 2

Were you able to assess all statistics in the manuscript, including the appropriateness of statistical tests used? No.

Were you able to directly test the methods? No.

Comments to author:

The paper presented by Dr. Ivancic et al. entitled "INSERT-seq enables high resolution mapping of genome integrated DNA using single molecule long read technologies" shows a new advanced strategy for the identification, analysis and definition of the genome integration sites were exogenous DNA fragments, provided using different platforms, are inserted. This is a very important and crucial step in the development of gene therapy strategies, either using addition strategies or gene editing ones. The manuscript clearly identifies this requirement and develop a pipeline where different on-the-market strategies are linked together to develop a more precise and accurate procedure. Moreover, the use of OxfordNanopore technology and the generation of longer reads clearly improves the presented strategy. However, there are important issues that need to be addressed before the manuscript being ready for publication in Genome Biology or elsewhere.

Major comments

In general, the manuscript has not been properly and carefully edited. There are supplementary figures that are cited but are not present in the manuscript. Supplementary Figures 3C, 6C, 6D and 8 are referred in the text but are not provided. Many abbreviations are referred but are not explained. What is the meaning of PB clones? What is the meaning of ONT platform? INSERTseq is the name the authors have develop for their new development. However, this is not clearly explained in the text. Clearly, the manuscript needs a review in depth to solve all

these issues. In the way it is now, it cannot be clearly understood.

Dfam database is mentioned many times but a clear explanation about it, apart from a reference in the bibliography is required.

The STAT-PCR is important in the pipeline presented. However it is only referred in methods. A brief explanation is required to understand why STAT-PCR is important/needed to get this improved procedure.

It is known that OxfordNanopore technology is more error prone than shorter sequencing technologies, such as Illumina based ones. This reviewer envisions that the inclusion of the UMI step could facilitate the resolution of this OxfordNanopore characteristic. A comparison in this aspect between Oxford Nanopore and Illumina based sequencers would be required to support the use of the Oxford Nanopore platform.

In the text a Piggybacktransposase-Cas9 fusion protein is mentioned. Figure 3 is in fact related to this fusion protein. However, nothing else is mentioned. A better explanation of this protein, potential references or additional details about the rationale and results obtained with this fusion protein would be required to understand its inclusion here and the objective of the presented figure

Minor comments

300 is highlighted in page 8 line 52, which probably would not be required

Reviewer 3

Were you able to assess all statistics in the manuscript, including the appropriateness of statistical tests used? Yes.

Were you able to directly test the methods? No.

Comments to author:

The authors present INSERT-seq, an Oxford Nanopore long-read sequencing based approach and coupled computational algorithm that aims to improve detection of DNA integration into edited cells. The focus is largely on improved detection within repetitive regions which represent a large fraction of the genome that is known to be poorly assessed by short-reads (e.g. Illumina) sequencing methods. This is an important problem to address and solutions would have considerable value. However, as presented, I am unable to fully assess the advance provided by the method. I detail some of my concerns/questions below:

Major

- Simulation study: Does the stated performance difference based on simulated reads hold even if many tens of millions (or more) short reads are generated? This is relevant since generating a short read can be considerably cheaper than a long read, and 1M reads is a tiny short-read sequencing run. The current simulation is a bit problematic since it generates an equal number of short vs long reads, resulting in 4X more sequenced bases for the long-read scenario. In practice, cost per base is considerably lower for short-read technologies so it would be more fair to have the short read scenario represent a much larger number of reads. I would ideally like to see detection rate curves as a function of total sequences bases for short vs long. (This may not be necessary if it can be shown that both cases saturate at a reasonably low read count.)

- How many reads were generated for the short-read (i.e. SHORT-eq) vs long-read INSERT experiments? The concern here is similar to the one in the previous point. To compare short vs long-reads you should have many fold more short reads in the comparison.
- The authors do not explicitly explain why the short-read approach yields lower detection rates. I presume it because of mappability issues in repetitive regions, where non-uniquely mapped reads are thrown out? This should be explored and explained.
- Can the unanchored peak calling approach not be applied equally well in the short-read scenario? I would guess that the loss of sensitivity of short-reads comes from non-uniquely mapped reads. If so, these should be mappable to Dfam which would enable detection of peaks in repeat regions, as is done for the long reads. If this is done, do we still see a sensitivity difference between long vs short reads? I would guess that the benefit of long reads is that some of these Dfam-mapped short read loci could potentially be assigned specific genomic locations. If so, this benefit should be stated more clearly.

Minor

- This may be a terminology issue, but I'm a bit unclear of the meaning of 'resolution' as it is used in the manuscript. I think of resolution as improving the accuracy with which an event is located (e.g. 1bp resolution vs 1kb resolution). I think it is being used to refer to improved sensitivity in this manuscript (i.e. we detect a site that is missed by other methods). If so, 'improved detection sensitivity' or something similar may be clearer to readers.
- The peak calling parameters (e.g. sd cutoff of 222.7823) are clearly very specific to this exact dataset. How do they vary when computed on a replicate (or subset) dataset?
- The mapping parameters (especially for the short-read analysis) should be specified. In particular, I would like to know how non-uniquely mapping reads are handled.

Reviewer #1: INSERT-seq enables high resolution mapping of genomically integrated DNA using single molecule long read technologies written by Dimitrije Ivancic and co-workers.

Viral vector mediated delivery is a method currently widely adopted in the field of human therapy applications. As recently examples were published of gene therapy trials based on such methods leading to undesired side effects the need for methods that can control for correct, but also undesired effects are of high interest.

This manuscript describes the development and initial validation of the INSERT-seq method. This method promises for the sensitive detection of LTR mediated DNA insertions at a-priory unknown positions and is, given the above, most relevant and timely.

Technically this method is based on the principle of STAT-seq/GUIDE-seq except here long read sequencing (LRS) is used yielding an increase in the number of uniquely identifiable integration sites. In addition, the authors adopted a mapping strategy designed to deal with integration sites yielding non-unique sequence reads.

The authors introduce INSERT-seq having performed a first set of validation experiments making it highly likely that their method outperforms STAT-PCR.

However, presented data provides insufficient body to support the claims as currently presented. Also, the manuscript looks to be written hastily leaving many incorrect referrals, missing legends figure numbers, performed but not described experiments making it difficult to read through in general. It will take serious effort correcting this. However, as most issues are textual, and few additional experiments are required I would recommend considering this manuscript for publication when corrections have been made.

Specific comments in order of appearance:

1. Title is too generic for the content of the manuscript

The title has been modified to “nanopore sequencing”

2. Due to placement of primer towards integration site in its current form only applicable to DNA that insert via known sequences like LTRs.

We envision that in most applications INSERT-seq the experimenter knows the end of the integrated DNA, moreover several primers could be added for each end. Nevertheless, non-canonical integration of only a portion of the payload, which is an occurring phenomenon in rAAV for example, can not be detected with INSERT-seq. This has been clarified in the manuscript with: “With the current protocol however, unbiased detection of partial payload integrations in which ends are missing, previously reported for rAAVs (Gil-Farina et al. 2016), remains unsolved.”

- b. The title states 'long read technologies' as in plural, except a single method was used for validation.

The title has been modified to “nanopore sequencing”

2. In figure 1a the authors aim to provide evidence for the benefit of using long read sequencing to increase the uniqueness of the identified integration site sequences. Hereto they use 100 random positions in the genome and make a comparison between a 250 and a 1000bp sequence at that position as described in the methods section. From that experiment INSERT-seq is claimed to provide a 4.2% benefit in the ability of uniquely calling integration site. This experiment raises three questions currently unaddressed

a. For this computational analysis to be relevant to the remainder of the manuscript, and as the use of long-read sequencing icw STAT-PCR is presented as novelty it is important to verify that the assumed 1kb read length matches the length of the experimental data. Aiming to verify this parameter from the current manuscript the majority of NGS reads are within 200bp from the insertion site (Figure 1c and supplemental figure 8) it will be good to address this properly. In addition, a Covaris G-tube was used to produce 2kb fragments, while these tubes are meant to produce 6-12kb fragments according to specifications. Currently it is not clear how the Covaris protocol was adjusted to yield 2kb fragments.

We have included supplementary figure 11 containing information regarding read length of covaris fragmented DNA. Since the library protocol of INSERTseq is amplification based, average read length decreases after library prep compared to fragmented DNA.

b. In figure 1b the finding of close to 40 new integrations in this dataset used to prepare figure 1a, of which in main text it is mentioned that 4.2% more sites are detected while in the methods this computational experiment was performed on 100 randomly selected sites. These number does not add up and suggest that more than 100 sites were used for this analysis. Please verify.

We obtained 100 insertions and performed the simulation 10 times, which gives a total number of 1000 insertions. We have specified the total number on the manuscript to improve clarity.

c. The total number of identified sites does not add up to 100 (in total or % is unclear from figure 1a), as at most 90 are found; all methods have their limitations and it is important to understand what these are; can the authors comment on what sites they did not identify and what was the cause?

We have detailed the undetected insertions (Supplementary table 3) and commented on them in the manuscript.

101 undetected insertions correspond to reads that map to multiple regions in the genome, which are discarded by a filtering step as the exact position can't be determined. 14 of those 101 insertions are detected when increasing read length to 5 kbp, suggesting that the sensitivity of the method increases when read length increases.

3. In both the abstract and discussion a resolution improvement of 10% is claimed as a general performance increase for INSERT-seq. Though the origin of the 10% is not explained, to my knowledge the only experiment showing nearing this figure is the experiment shown in figure 1f. This single experiment covering ~40 integrations is however not sufficient to make such a broad claim. For example, from the computational experiment the increase is shown to be 4.2% which is quite a different range and interrogating many more genomic locations.

We have modified the percentage of improvement based on the new analysis of the MOPO sample, where we find a 7.3% of new integrations.

We have commented the importance of the different vectors of integration, which can affect the percentage of detected insertions based on their differential integration site preferences.

4. For the purpose the authors generated the MOPO control cell line to be used as a reference. In the main text pg 3 line ~15 it states that 42 viral insertions should be present. Supplemental figure 2 however shows 37 copies to be present. That finding makes that the reference value looks more alike the short-read finding which is presented as being incomplete. I'm afraid that with this outcome of the ddPCR (here used as the independent reference method) the authors should make an extra effort on showing the additional sites are indeed true positives as now presented.

We have performed several repeats of the copy number quantification using ddPCR (Supplementary Figure 2). We quantified at different dilutions to rule out issues with signal

saturation in the method (Supplementary Figure 2a), however, estimated CN varied between 21 and 45.5 for the different dilutions, indicating substantial error in the quantification. In order to validate the insertions, we performed INSERTseq on the 3' side of the payload and an additional primer pair at the 5' of the lentiviral payload.

The same results were obtained with both 5' end replicates. We detected two new integration sites when mapping the 3' end and two sites detected with the 5' end were not detected with the 3' end.

The insertions found only with the 3' end mapping correspond to an insertion at chr10 falling in an LTR (MLT1C2, which was detected by the unanchored peak calling method) and one at chr22 falling in a region with multiple repeats classified as unknown in the Dfam database. The two insertions found only with the 5' end mapping correspond to the insertion at chr7 which was also not detected with SHORT-seq and an insertion at chr1 falling in a SINE (AlISz6).

Supplementary Figure 2 | CN determination

5. While introducing a new quality control method no replicate experiments are shown. The MOPO and possibly the 100 clones PB1 sample would make good candidates.

We repeated INSERTseq on the MOPO sample with a new primer set and the other side of the payload (3' end).

We were able to map the same integration sites with both repeats mapping the 5' end. Two of those insertions were not found when mapping the 3' end and two new integration sites were found with 3' end primers.

6. Most of the data presented in figure 1 d and e is not described in the main text.

We have added description:

“In the analyzed MOPO sample, the insertions found in repetitive regions comprised LINES, SINEs, LTR retrotransposons and DNA transposons (Fig. 1d) and insertions were happening

preferentially in intronic regions with a higher percentage than the randomly distributed model sample (Fig. 1e) ”

Both figures are now cited and commented in the section “Read length dependency” of the manuscript.

7. When describing the limit of detection it is currently not clear how this number was obtained. In supplemental figure 3 coverage and DHD are plotted vs proportion positive sample (%), but how many reads were used or how peak calling was executed including significance cut-of percentages and how the increased number of false insertion sites is treated in these experiments is not explained.

We have improved the explanation about how the LOD is obtained in the section “Overview of library prep implementation and optimization” of the manuscript.

We analyzed a mono-clonal cell line with one targeted insertion by performing serial dilutions of the same sample at 1:100, 1:1000 and 1:10000. The detection of the target integration site at the dilution 1:1000 and not at the dilution 1:10000 determines INSERT-seq limit of detection of 0.1%.

We specified the sequencing output obtained per sample, being 300000 raw reads for INSERT-seq and 30000 after UMI clustering and 230000 reads for SHORT-seq after UMI clustering.

We improved the explanation about false positives due to highly diluted samples. We have modified Supplementary figure 3 to improve clarity. Most of those false positive insertions were found to come from the cross contamination with the MOPO sample, which highlights the importance of sample tracing, specially in low enriched samples.

8. Supplemental figure 6c is difficult to understand without figure legends or explanation.

We have added figure legends.

9. Out of the 4 new sites presented in figure 1c. It is not clear
a. why the top 2 integration sites do not show any reads using short read-seq while the bottom two show very high coverage and
b. having such high coverage, why are the bottom two peaks missed in peak calling?

We have corrected and further clarified the explanation. The last insertion of figure 1c corresponded to a true positive insertion detected with both methods, shown as an example, we have removed it to improve clarity.

10. In general it was difficult to read through the manuscript due to
a. the lack of page numbers, (for reference below, pg1 is considered the title/abstract pg)
We have added page numbers
b. the lack of figure numbers
We have added figure numbers

c. the lack of figure legends.

We have added figure legends

d. Pg1 line 40; 'in-vivo' where ex-vivo is the more correct description.

We have modified experimental by ex-vivo to clarify. We include in-vivo samples (rAAV-8 integration in mice liver)

e. Pg2 line 24: STAT-PCR [23] was called GUIDE-seq by the authors of the original publication

We have modified STAT-PCR by GUIDE-seq when referring to the original methodology.

f. Pg2 line 44: no explanation of 'PB' which can easily be done in line 23 of the same document

Added clarification of what PB is referring to.

g. Pg 3 line 8-9 unclear what is meant with '..we detect a high number of repeats (supplementary Fig. 1.)' How does the sequence length impact on the length of a repeat?

We clarified this explanation, we examined the length of the repeats, independent of any read length.

"examining the length of all repetitive regions from the human genome annotated in Dfam database [34], we detect a high number of repeats longer than 500 bp (Supplementary Fig. 1)."

h. Pg3 line 11 or 12 (line numbering does not align with the actual text) it states '... in the optimal range predicted by the model.' However, the mentioned model is not explained nor introduced.

We corrected this statement, the optimal range was obtained from the observation to all repetitive regions in the human genome.

i. The y-axis on figure 2e top panel is incomplete

We fixed this issue.

j. pg3 line 50. It is not clear where the referral of '...one overrepresented insertion....(Fig. 2e). refers to when looking at figure 2e.

We have clarified this point.

k. pg3 line 60 (bottom line) suppl figure 3c does not exist, please refer to correct figure

We have fixed this issue.

l. pg4 line 19: mentioning of 'a clinically relevant sample' followed by 'liver tissue from mice' raises a question mark. Though one can argue that such an animal experiment is related to the direct medical treatment or testing of patients, but it reads artificial/farfetched as the relationship is indirect at most. Rephrasing this sentence would help to increase credibility.

The text was replaced by "an in-vivo model" which better reflects the nature of the sample, as suggested.

Reviewer #2: The paper presented by Dr. Ivancic et al. entitled "INSERT-seq enables high resolution mapping of genome integrated DNA using single molecule long read technologies"

shows a new advanced strategy for the identification, analysis and definition of the genome integration sites were exogenous DNA fragments, provided using different platforms, are inserted. This is a very important and crucial step in the development of gene therapy strategies, either using addition strategies or gene editing ones. The manuscript clearly identifies this requirement and develop a pipeline where different on-the-market strategies are linked together to develop a more precise and accurate procedure. Moreover, the use of OxfordNanopore technology and the generation of longer reads clearly improves the presented strategy. However, there are important issues that need to be addressed before the manuscript being ready for publication in Genome Biology or elsewhere.

Major comments

In general, the manuscript has not been properly and carefully edited. There are supplementary figures that are cited but are not present in the manuscript. Supplementary Figures 3C, 6C, 6D and 8 are referred in the text but are not provided.

We have modified figure citations.

Many abbreviations are referred but are not explained. What is the meaning of PB clones? What is the meaning of ONT platform?

We have clarified

“population of 100 clones edited with HypB transposase (referred to as PB clones)”

“Oxford Nanopore Technologies (ONT) ”

INSERTseq is the name the authors have develop for their new development. However, this is not clearly explained in the text.

We have clarified

“To study the effect of repetitive elements on capturing insertion sites, we implemented a model that showed significant dependency of read length for accurately resolving insertion sites. Next, we implemented a series of steps in the library prep protocol and analysis pipeline to enable efficient single-tail adapter/tag (STAT-PCR) based long read sequencing to capture insertion sites across the genome, that we named INSERT-seq.”

Clearly, the manuscript needs a review in depth to solve all these issues. In the way it is now, it cannot be clearly understood. Dfam database is mentioned many times but a clear explanation about it, apart from a reference in the bibliography is required.

We have described the database.

“Furthermore, examining the length of all repetitive regions from the human genome annotated in Dfam database [34]”

The STAT-PCR is important in the pipeline presented. However it is only referred in methods. A brief explanation is required to understand why STAT-PCR in important/needed to get this improved procedure.

We have added more details regarding STAT-PCR

“single-tail adapter/tag (STAT-PCR)”

“STAT-PCR single tailed adaptor ensures selective amplification of fragments containing both adaptor and vector sequence, since the primer targeting the adaptor region does not bind to the single stranded version of the adaptor and it only binds when amplification from the primer targeting the vector occurs”

It is known that OxfordNanopore technology is more error prone than shorter sequencing technologies, such as Illumina based ones. This reviewer envisions that the inclusion of the UMI step could facilitate the resolution of this OxfordNanopore characteristic. A comparison in this aspect between Oxford Nanopore and Illumina based sequencers would be required to support the use of the Oxford Nanopore platform.

We have calculated the error rate of mapped reads for long and short reads.

Info was added:

“We analyzed the impact of sequencing output to detecting integration sites (Supplementary Fig. 9c). We found that there was no increase in detection upon increasing bp output starting from 250Mbp to 1.5Gbp for the simulated insertions.”

In the text a Piggybacktransposase-Cas9 fusion protein is mentioned. Figure 3 is in fact related to this fusion protein. However, nothing else is mentioned. A better explanation of this protein, potential references or additional details about the rationale and results obtained with this fusion protein would be required to understand its inclusion here and the objective of the presented figure

We have included citation to our recent work describing Piggybacktransposase-Cas9 (not published at the time of submission of this manuscript). We added clarification of the chimera. (Pallarès-Masmitjà et al. 2021)

Minor comments

300 is highlighted in page 8 line 52, which probably would not be required

We fixed this issue

Reviewer #3: The authors present INSERT-seq, an Oxford Nanopore long-read sequencing based approach and coupled computational algorithm that aims to improve detection of DNA integration into edited cells. The focus is largely on improved detection within repetitive regions which represent a large fraction of the genome that is known to be poorly assessed by short-reads (e.g. Illumina) sequencing methods. This is an important problem to address and solutions would have considerable value. However, as presented, I am unable to fully assess the advance provided by the method. I detail some of my concerns/questions below:

Major

- Simulation study: Does the stated performance difference based on simulated reads hold even if many tens of millions (or more) short reads are generated? This is relevant since generating a short read can be considerably cheaper than a long read, and 1M reads is a tiny short-read sequencing run. The current simulation is a bit problematic since it generates an equal number of short vs long reads, resulting in 4X more sequenced bases for the long-read scenario. In practice, cost per base is considerably lower for short-read technologies so it would be more fair to have the short read scenario represent a much larger number of reads. I would ideally like to see detection rate curves as a function of total sequenced bases for short vs long. (This may not be necessary if it can be shown that both cases saturate at a reasonably low read count.)

- How many reads were generated for the short-read (i.e. SHORT-seq) vs long-read INSERT experiments? The concern here is similar to the one in the previous point. To compare short vs long-reads you should have many fold more short reads in the comparison.

We have compared the increase in detected insertions when increasing the number of sequenced bases in the computational model (Supplementary fig. 9c) where we didn't find

differences. A comparison with higher short read count of the MOPO sample was made (Supplementary Fig. 9a) where one insertion at chr7 is recovered (Supplementary Fig. 9b).

- The authors do not explicitly explain why the short-read approach yields lower detection rates. I presume it because of mappability issues in repetitive regions, where non-uniquely mapped reads are thrown out? This should be explored and explained.

We have further explained the filtering process where we discard reads that map to multiple regions or reads that map with a map quality below 30.

- Can the unanchored peak calling approach not be applied equally well in the short-read scenario? I would guess that the loss of sensitivity of short-reads comes from non-uniquely mapped reads. If so, these should be mappable to Dfam which would enable detection of peaks in repeat regions, as is done for the long reads. If this is done, do we still see a sensitivity difference between long vs short reads? I would guess that the benefit of long reads is that some of these Dfam-mapped short read loci could potentially be assigned specific genomic locations. If so, this benefit should be stated more clearly.

We have applied the unanchored peak calling algorithm to short reads where we detect two unanchored insertions.

Minor

- This may be a terminology issue, but I'm a bit unclear of the meaning of 'resolution' as it is used in the manuscript. I think of resolution as improving the accuracy with which an event is located (e.g. 1bp resolution vs 1kbp resolution). I think it is being used to refer to improved sensitivity in this manuscript (i.e. we detect a site that is missed by other methods). If so, 'improved detection sensitivity' or something similar may be clearer to readers.

We have replaced resolution term when referring to improved detection sensitivity, to improve the clarity of the text.

- The peak calling parameters (e.g. sd cutoff of 222.7823) are clearly very specific to this exact dataset. How do they vary when computed on a replicate (or subset) dataset?

We thank the reviewer for raising this point. We corrected the peak calling by fitting the coverage distribution of peaks to a beta distribution. Such fitting is not dependent on read length and can be applied to long-read and short-read samples and to unanchored peak calling. We have added replicates from the MOPO sample to add consistency.

- The mapping parameters (especially for the short-read analysis) should be specified. In particular, I would like to know how non-uniquely mapping reads are handled.

We have specified mapping parameters. The default parameters "map-ont" were used for mapping of long reads and the default parameters "sr" and "--secondary=yes" were used for mapping of short reads.

Second round of review

Reviewer 1

I'd like to start with complimenting the authors on the adjustments made, they significantly improved the readability of the manuscript.

Doing so, far most of the points raised have been addressed, however a few critical points remain and are outlined below.

1. In figure 1a the authors aim to provide evidence for the benefit of using long read sequencing to increase the uniqueness of the identified integration site sequences. Hereto they use 100 random positions in the genome and make a comparison between a 250 and a 1000bp sequence at that position as described in the methods section. From that experiment INSERT-seq is claimed to provide a 4.2% benefit in the ability of uniquely calling integration site. This experiment raises three questions currently unaddressed

- a. For this computational analysis to be relevant to the remainder of the manuscript, and as the use of long-read sequencing icw STAT-PCR is presented as novelty it is important to verify that the assumed 1kb read length matches the length of the experimental data. Aiming to verify this parameter from the current manuscript the majority of NGS reads are within 200bp from the insertion site (Figure 1c and supplemental figure 8) it will be good to address this properly. In addition, a Covaris G-tube was used to produce 2kb fragments, while these tubes are meant to produce 6-12kb fragments according to specifications. Currently it is not clear how the Covaris protocol was adjusted to yield 2kb fragments.

To address this question the authors have included supplemental figure 11. That additional data is highly informative but has two issues

- **Minor, this supplemental figure is not referred to in the main text**
- **Major, sup fig 11 c shows the average read size of INSERT-seq to be ~250-500bp as far as I can judge matching the afore mentioned length seen in former supplemental figure 8 (now sup fig 6). As mentioned above, this fact makes that the theoretical advantage calculated comparing 250 and 1000bp inserts does not apply for INSERT-seq as it now reads from the manuscript. To make this analysis relevant the authors should used similar sized fragments for their computational analysis as the size in the INSERT-seq method.**

2. In both the abstract and discussion a resolution improvement of 10% is claimed as a general performance increase for INSERT-seq. Though the origin of the 10% is not explained, to my knowledge the only experiment showing nearing this figure is the experiment shown in figure 1f. This single experiment covering ~40 integrations is however not sufficient to make such a broad claim. For example, from the computational experiment the increase is shown to be 4.2% which is quite a different range and interrogating many more genomic locations.

The authors have recalculated this value and now state it to be 7.3%. Though now explained how this value was calculated (3 new sites of 41 total sites, figure 1), the mentioning in supplemental figure 9 that using more short reads one of the 3 missing sites was now also detected this number should be

adjusted. With this finding now 2 new sites are uniquely detected using INSERT-seq which resembles 4.8% in this example. The authors should adjust this calculation or better justify why they think 7.3% is the correct number to state.

3. For the purpose the authors generated the MOPO control cell line to be used as a reference. In the main text pg 3 line ~15 it states that 42 viral insertions should be present. Supplemental figure 2 however shows 37 copies to be present. That finding makes that the reference value looks more alike the short-read finding which is presented as being incomplete. I'm afraid that with this outcome of the ddPCR (here used as the independent reference method) the authors should make an extra effort on showing the additional sites are indeed true positives as now presented.

To address this question the authors are showing additional ddPCR reactions using 1x and 0.2x diluted samples. From supplemental figure 2 where this data is presented the authors are correct in stating that this analysis suffers from substantial error in the quantification. Indeed variation is substantial, but most variation seems to originate from the 0.2x diluted samples, all other datapoints look to average around 30 copies.

In the manuscript the authors state to have identified 43 copies (sup fig 10, both 3' and 5' datasets combined). In addition, they claim to have identified 10 unique non-anchored integration sites which brings the total of sites to 53. This finding makes the deviation from the ddPCR finding even larger.

However, as the authors show consistent results in their replicate experiments, in their results using the different primers I trust the INSERT-seq data as presented in supplemental figure 9 and 10 is more reliable than the ddPCR.

However, for the additional 10 identified sites in the non-anchored regions such data is not presented and does cause the number of copies to deviate substantially (much more than the normal 10% known from ddPCR) which does create doubt on the correctness of this part of the analysis. Using ddPCR as the gold standard in their experiments it would improve the manuscript if the authors would be able to address this deviation properly.

4. When describing the limit of detection it is currently not clear how this number was obtained. In supplemental figure 3 coverage and DHD are plotted vs proportion positive sample (%), but how many reads were used or how peak calling was executed including significance cut-of percentages and how the increased number of false insertion sites is treated in these experiments is not explained.

This point has been more clearly explained which is much appreciated. However, inconsistencies remain

- In the rebuttal text it is written that 300.000 reads are used, in the methods section it states 500,000 reads to be used for the 1:100 and the 1:1000 and 150,000 reads were used for the 1:10.000 dilution.
- Dilution samples have been analyzed starting from different read numbers
- It is now mentioned that false positives are identified, but the number of false positives is not mentioned.

- There does not seem to be a linear correlation between the number of reads and the dilution, both the 1:100 and 1:000 samples look to have comparable integration site reads. How do the number of reads from the true positive compare to the false positives?
- It is stated that the majority of false positives are caused by cross contamination from another sample. With the absence of a linear correlation mentioned above, can the authors exclude the possibility that the true positive reads also originate from cross contamination?
- Realizing that the LOD is calculated based on a single experiment containing a single datapoint which in it's totality is suffering from cross-contamination, did the authors consider to repeat it?

5. In general it was difficult to read through the manuscript due to

- The y-axis on figure 2e top panel remains incomplete
- Quite a few typo's throughout remain
- Supplemental figure 4d as mentioned in the main text on pg3 is not present

1. In figure 1a the authors aim to provide evidence for the benefit of using long read sequencing to increase the uniqueness of the identified integration site sequences. Hereto they use 100 random positions in the genome and make a comparison between a 250 and a 1000bp sequence at that position as described in the methods section. From that experiment INSERT-seq is claimed to provide a 4.2% benefit in the ability of uniquely calling integration site. This experiment raises three questions currently unaddressed

a. For this computational analysis to be relevant to the remainder of the manuscript, and as the use of long-read sequencing icw STAT-PCR is presented as novelty it is important to verify that the assumed 1kb read length matches the length of the experimental data. Aiming to verify this parameter from the current manuscript the majority of NGS reads are within 200bp from the insertion site (Figure 1c and supplemental figure 8) it will be good to address this properly. In addition, a Covaris G-tube was used to produce 2kb fragments, while these tubes are meant to produce 6-12kb fragments according to specifications. Currently it is not clear how the Covaris protocol was adjusted to yield 2kb fragments.

To address this question the authors have included supplemental figure 11. That additional data is highly informative but has two issues

- Minor, this supplemental figure is not referred to in the main text
- Major, sup fig 11 c shows the average read size of INSERT-seq to be ~250-500bp as far as I can judge matching the afore mentioned length seen in former supplemental figure 8 (now sup fig 6). As mentioned above, this fact makes that the theoretical advantage calculated comparing 250 and 1000bp inserts does not apply for INSERT-seq as it now reads from the manuscript. To make this analysis relevant the authors should used similar sized fragments for their computational analysis as the size in the INSERT-seq method. Figure referenced and explained in section "Read length dependency" from the manuscript. The model was repeated with read lengths of 100 and 350 after the detection of a shorter length of mapped reads:

"Since the library protocol of INSERTseq is amplification based, the average read length decreases after library prep compared to fragmented DNA. Mapped read length of long reads ranges from 41 to 5718bp with a mean of 329bp while mapped length of short reads range from 25 to 239bp with a mean of 109bp in the analysed MOPO sample (Supplementary Fig. 11). The increase of detected insertions was analysed with modelled reads of 100 (short) and 350 (long) base pairs, finding a not statistically significant increase of 2% (Supplementary Fig. 9e). Such results confirm that the potential of INSERTseq improves with longer reads."

Supplementary Fig. 9e, Supplementary. Fig. 11d and legend were added.

e) Number of true positive (TP) insertions detected with read length of 100 and 350 base pairs. Statistically significant differences ($p < 0.05$, ns = not significant).

c) Density distribution of mapped read length of the MOPO sample. Mapped length of long reads range from 41 to 5718 with a mean of 329 while mapped length of short reads range from 25 to 239 with a mean of 109. **d)** Close view of the density distribution of mapped read length of the MOPO sample. With read length between 0-350.

2. In both the abstract and discussion a resolution improvement of 10% is claimed as a general performance increase for INSERT-seq. Though the origin of the 10% is not explained, to my knowledge the only experiment showing nearing this figure is the experiment shown in figure 1f. This single experiment covering ~40 integrations is however not sufficient to make such a broad claim. For example, from the computational experiment the increase is shown to be 4.2% which is quite a different range and interrogating many more genomic locations.

The authors have recalculated this value and now state it to be 7.3%. Though now explained how this value was calculated (3 new sites of 41 total sites, figure 1), the mentioning in supplemental figure 9 that using more short reads one of the 3 missing sites was now also detected this number should be adjusted. With this finding now 2 new sites are uniquely detected using INSERT-seq which resembles 4.8% in this example. The authors should adjust this calculation or better justify why they think 7.3% is the correct number to state.

We modified the statements by providing a range between 4.8 and 7.3%. The calculation of a 7.3% increase coming from the detection of 3 new insertions and 4.8% from 2 new insertions is explained in section “Read length dependency”

3. For the purpose the authors generated the MOPO control cell line to be used as a reference. In the main text pg 3 line ~15 it states that 42 viral insertions should be present. Supplemental figure 2 however shows 37 copies to be present. That finding makes that the reference value looks more alike the short-read finding which is presented as being incomplete. I'm afraid that with this outcome of the ddPCR (here used as the independent reference method) the authors should make an extra effort on showing the additional sites are indeed true positives as now presented.

To address this question the authors are showing additional ddPCR reactions using 1x and 0.2x diluted samples. From supplemental figure 2 where this data is presented the authors are correct in stating that this analysis suffers from substantial error in the quantification. Indeed variation is substantial, but most variation seems to originate from the 0.2x diluted samples, all other datapoints look to average around 30 copies.

In the manuscript the authors state to have identified 43 copies (sup fig 10, both 3' and 5' datasets combined). In addition, they claim to have identified 10 unique non-anchored integration sites which brings the total of sites to 53. This finding makes the deviation from the ddPCR finding even larger.

However, as the authors show consistent results in their replicate experiments, in their results using the different primers I trust the INSERT-seq data as presented in supplemental figure 9 and 10 is more reliable than the ddPCR.

However, for the additional 10 identified sites in the non-anchored regions such data is not presented and does cause the number of copies to deviate substantially (much more than the normal 10% known from ddPCR) which does create doubt on the correctness of this part of the analysis. Using ddPCR as the gold standard in their experiments it would improve the manuscript if the authors would be able to address this deviation properly.

We have clarified that unanchored peaks are putative peaks: "However, due to the technical challenges that repetitive regions present, unanchored peaks are only reported as putative peaks." Thus, not taken into account when stating the resolution improvement of INSERTseq.

4. When describing the limit of detection it is currently not clear how this number was obtained. In supplemental figure 3 coverage and DHD are plotted vs proportion positive sample (%), but how many reads were used or how peak calling was executed including significance cut-off percentages and how the increased number of false insertion sites is treated in these experiments is not explained.

This point has been more clearly explained which is much appreciated. However, inconsistencies remain

- In the rebuttal text it is written that 300,000 reads are used, in the methods section it states 500,000 reads to be used for the 1:100 and the 1:1000 and 150,000 reads were used for the 1:10,000 dilution.
- Dilution samples have been analyzed starting from different read numbers
- It is now mentioned that false positives are identified, but the number of false positives is not mentioned.
- There does not seem to be a linear correlation between the number of reads and the dilution, both the 1:100 and 1:1000 samples look to have comparable integration site reads. How do the number of reads from the true positive compare to the false positives?

- It is stated that the majority of false positives are caused by cross contamination from another sample. With the absence of a linear correlation mentioned above, can the authors exclude the possibility that the true positive reads also originate from cross contamination?
- Realizing that the LOD is calculated based on a single experiment containing a single datapoint which in it's totality is suffering from cross-contamination, did the authors consider to repeat it?

We have repeated the LOD detection experiment. We mixed serial dilutions of a monoclonal cell line (MN2, insertion at chr 12, red bar), with a constant amount of MN7 (insertion at chr 6, blue bar). Insertion at chr 6 acts as a reference, and insertion at chr12 is serially diluted, in the upper plot. We performed the same experiment using different reference monoclonals and dilution to confirm the observations (lower plot, MN2, with insertion at chr 12 as reference, and MN 10. With two insertions at chr3 and chr9, serially diluted). In both cases detection threshold is at 1:100.

5. In general it was difficult to read through the manuscript due to
 - o The y-axis on figure 2e top panel remains incomplete: it is complete
 - o Quite a few typo's throughout remain
 - o Supplemental figure 4d as mentioned in the main text on pg3 is not present: modified to 4c

Third round of review

Reviewer 1

Following the 2nd revision all major concerns have been addressed.

Four minor items remain following the revision:

1. LOD has been corrected to 1% throughout the manuscript, except 0.1% is stated in the abstract.
2. Pg 3 now mentions that a high error rate was found in the ddPCR experiments yielding a CN ranging from 21-45.5 which the authors claim to fall within range of the number of insertion sites detected by the method (41). However, the number of integration sites mentioned did not take into account the 10 putative sites which the authors say to exclude in the novel sites detected, but are being appreciated further on in the manuscript. This raises 2 concerns. A. if the putative sites are considered relevant then the total number of integration sites is $41+10=51$ and should be discussed as such. B. the ddPCR that was used as an independent method for determining the actual nr of vector copies present is clearly not yielding robust results. With this much deviation present, well above the generally accepted 10% deviation, to me it looks like this assay did not work properly and wonder if mentioning this analysis in its current form hold any value? Maybe the authors could strengthen their point by focusing the validation on the use of the 5' and 3' data that due to the setup (use of different primer, detection of different fusion read sequence) for me are much more reliable and informative.
3. Pg 4 mentions 'SOHRT-seq data' where likely 'SHORT-seq data' is intended
4. In the methods section covering library preparation and sequencing it states that DNA is fragmented to ~66kbp where from the manuscript I got the impression that 66kbp was the length of DNA after isolation before fragmentation was applied. Please check.

Authors' response

1. LOD has been corrected to 1% throughout the manuscript, except 0.1% is stated in the abstract.

This has been corrected in the current version of the abstract

2. Pg 3 now mentions that a high error rate was found in the ddPCR experiments yielding a CN ranging from 21-45.5 which the authors claim to fall within range of the number of insertion sites detected by the method (41). However, the number of integration sites mentioned did not take into account the 10 putative sites which the authors say to exclude in the novel sites detected, but are being appreciated further on in the manuscript. This raises 2 concerns. A. if the putative sites are considered relevant then the total number of integration sites is $41+10=51$ and should be discussed as such. B. the ddPCR that was used as an independent method for determining the actual nr of vector copies present is clearly not yielding robust results. With this much deviation present, well above the generally accepted 10% deviation, to me it looks like this assay did not work properly and wonder if mentioning this analysis in its current form hold any value? Maybe the authors could strengthen their point by focusing the validation on the use of the 5' and 3' data that due to the setup (use of different primer, detection of different fusion read sequence) for me are much more reliable and informative.

We agree that ddPCR is not much informative at this point and we removed this piece of data, relying on the two sided validation of the insertions

3. Pg 4 mentions 'SOHRT-seq data' where likely 'SHORT-seq data' is intended

Corrected

4. In the methods section covering library preparation and sequencing it states that DNA is fragmented to ~66kbp where from the manuscript I got the impression that 66kbp was the length of DNA after isolation before fragmentation was applied. Please check.

We meant 6kb. Thanks and corrected.